# Antithrombotic drugs for cardiovascular risk reduction in patients with lower limb peripheral arterial disease: protocol for a systematic review and network meta-analysis of randomised controlled trials

David Brooke Sidebottom [1,2] Chao Huang,[3] Daniel Carradice,[3,4] Peter J Holt,[1,2] Karen John-Pierre,[5] Iain Nicholas Roy [1,2]

For numbered affiliations see end of article.

**Correspondence to**
Mr Iain Nicholas Roy;
iroy@sgul.ac.uk

## ABSTRACT

**Introduction** The optimal antithrombotic regimen to reduce the risk of vascular events in patients with peripheral arterial disease (PAD) is contentious. This systematic review and network meta-analysis (NMA) aims to define the relative efficacy and risks of previously investigated antithrombotic medication regimens in preventing major cardiovascular events, vascular limb events and mortality in patients with PAD.

**Methods and analysis** A peer-reviewed, systematic search will be executed in English on Medline, Embase, Cochrane (CENTRAL), Web of Science and Google Scholar databases in late 2022. The WHO International Clinical Trials Registry platform will also be searched for ongoing trials. Abstracts will be screened independently by two researchers for randomised controlled trials meeting the review criteria. All associated publications including the study protocol will be sought and evaluated together against prespecified inclusion/exclusion criteria. Two researchers will extract the data into a prepiloted extraction form. Risk-of-bias assessments will be performed using the Cochrane 'Risk-of-Bias V.2' criteria by individuals with domain expertise. All differences will be resolved by consensus or a third individual for ties. Included trials will be summarised. An NMA will be performed, subject to checks of assumptions. Both primary and secondary outcomes will be analysed on a whole network basis. Pairwise comparisons and league tables will be produced. Prespecified subgroup analyses will include sex, ethnicity, disease status, conservative versus interventional management and key comorbidities. The findings will be evaluated using the Grading of Recommendation Assessment, Development and Evaluation, informed by patient and public involvement work.

**Ethics and dissemination** This is a systematic review of data in the public domain and does not require ethical approval. Dissemination will include presentations to key vascular and patient organisations, publication in a peer-reviewed journal and an open-access repository of the study data.

## STRENGTHS AND LIMITATIONS OF THIS STUDY

⇒ A broad highly sensitive search strategy has been designed in collaboration with an experienced librarian to facilitate the most comprehensive review of this topic area in existence.

⇒ This review and its aims were closely informed by people with lived experience of peripheral arterial disease, with further collaborative work with patients planned during the analysis and interpretation phases.

⇒ A comprehensive series of subgroup analyses will provide the granularity necessary to understand potential differences between various antithrombotic combinations for key patient groups.

⇒ The number of possible combinations of antithrombotic agents and doses is large and will require a pragmatic combining of these data.

⇒ A number of robust sensitivity analyses are prespecified to explore anticipated high heterogeneity between studies, related to the inclusion/exclusion criteria of the available literature.

**PROSPERO registration number** CRD42023389262.

## INTRODUCTION

Peripheral arterial disease (PAD) describes a narrowing or occlusion of the peripheral arteries, which typically affects the lower limbs and is most often caused by atherosclerosis or atherothrombosis.[1] PAD may be asymptomatic or cause symptoms such as intermittent claudication (IC), where diminished circulation leads to pain in the lower limb on exertion that is relieved by rest.[2] More severe PAD can result in chronic limb-threatening ischaemia (CLTI), a clinical pattern representing threatened limb viability. CLTI is characterised by chronic, inadequate tissue perfusion

at rest and is defined by ischaemic rest pain and/or tissue loss.[3 4] In the UK, around 2.2 million people have some degree of PAD, while the prevalence rises with age to 16% in those aged over 70 years.[2 5–7] Between 10% and 30% of individuals with PAD experience IC,[3] while the population prevalence of reported CLTI is between 0.4% and 2%.[3 4] Other risk factors for both the development and progression of PAD include male gender, smoking, hypertension, diabetes, dyslipidaemia and obesity.[3 5]

The direct consequences of PAD include impaired quality of life,[2 8] psychosocial sequelae,[9] tissue loss (ulceration and gangrene) in CLTI,[3] amputation[10] and procedural complications resulting from invasive treatments.[2] Patients with PAD are three times more likely to die of cardiovascular causes including myocardial infarction, stroke and suffer from vascular dementia, renovascular disease and mesenteric disease.[1 11 12] Moreover, 10%–15% of patients presenting with IC die of cardiovascular causes over the following 5 years, while 20% experience non-fatal cardiovascular events.[8] Therefore, guidelines recommend the assessment of cardiovascular risk, and management of all key modifiable risk factors such as smoking, glycaemic control, dyslipidaemia, hypertension, body weight and exercise levels.[3 13]

Antithrombotic pharmacological therapy, which may involve one or more antiplatelet and/or anticoagulant agents, has been repeatedly demonstrated in randomised controlled trials (RCTs) to reduce occlusive vascular events or their recurrence.[14–22] However, the optimal antithrombotic regimen for the management of PAD is contentious, both overall and in key subpopulations such as those with diabetes. Multiple RCTs compared the investigational product to aspirin monotherapy,[14–16 19] which is not the recommended agent for the medical management of symptomatic PAD.[3 23] Furthermore, guidance from the National Institute for Health and Care Excellence (NICE) is incongruous; NICE Technology appraisal guidance 210 states "Clopidogrel is recommended as an option to prevent occlusive vascular events for people who have … PAD or multivascular disease",[23] while NICE Technology appraisal guidance 607 states "Rivaroxaban plus aspirin is recommended … as an option for preventing atherothrombotic events in adults with coronary artery disease or symptomatic PAD who are at high risk of ischaemic events".[24] Antithrombotic therapy for patients undergoing interventions such as surgical bypass or endovascular intervention has separate recommendations.

The current guidance on the medical management of PAD generates confusion for both clinicians and patients, while variation in clinical practice may cause harm. No published RCT has directly compared the two NICE recommended antithrombotic regimens, clopidogrel monotherapy and rivaroxaban plus aspirin, in patients with PAD. Furthermore, there is no RCT comparison of these antithrombotic medications in patients with PAD either underway or being planned. Therefore, this network meta-analysis (NMA) aims to address a critical unanswered question in daily clinical practice and

understand the relative efficacy and risks associated with each antithrombotic regimen in preventing cardiovascular events, limb loss, death and key adverse events (AEs) in different patient groups with PAD.

## AIMS

To understand the relative efficacy and safety of previously investigated antithrombotic medications in preventing major cardiovascular events, limb loss, mortality and bleeding events in patients with symptomatic lower limb PAD. A secondary aim is to investigate whether different patient groups have superior outcomes from different antithrombotic regimens.

## METHOD

The full protocol is available online,[25] but in summary, a systematic review and NMA of antithrombotic agents for patients with PAD will be undertaken. The review process will run from November 2022 to October 2023. The review is registered with the PROSPERO database,[26] and will be reported in line with the latest Preferred Reporting Items for Systematic Reviews and Meta-Analyses guidelines and relevant extensions for search strategies and NMAs.[27–29]

### Types of studies

All published and available unpublished RCT trial results will be included. All other study designs, including crossover study designs and non-randomised studies, will be excluded.

### Types of participants

People with PAD will be eligible, defined as (1) symptoms and a diagnosis of PAD by a clinician with experience in PAD and/or (2) a previous procedure to treat PAD (revascularisation procedure or amputation) and/or (3) objective evidence of lower limb arterial malperfusion. The Ankle Brachial Pressure Index (ABPI) is the most common objective measure of lower limb perfusion. Differing definitions of PAD exist using ABPI; for the avoidance of doubt, an ABPI of ≤0.9 will be used as objective evidence of malperfusion.

No limits will be set on age, country or previous therapy.

It is known that trials relevant to this review included patients with a variety of atherosclerotic disease phenotypes (eg, coronary heart disease and/or ischaemic stroke in addition to PAD). Additionally, some trials are known to have included patients with carotid artery atherosclerotic disease within the PAD subgroup. These will be included provided that (1) a defined set of patients with symptomatic PAD was included, (2) one or more outcomes was specifically reported for the PAD subgroup and (3) patients without lower limb PAD do not comprise greater than 25% of the PAD subgroup but do have some other form of atherosclerotic pathology.

## Types of interventions

Trials comparing one antithrombotic regimen to another, or to placebo, will be eligible for inclusion. Antithrombotic medication regimen will be defined as any individual or combination of medications listed in the British National Formulary as an antiplatelet, anticoagulant, glycoprotein IIb/IIIa inhibitor (online supplemental material) and/or any other individual or combination of medications reporting to inhibit platelets or fibrin aggregation in thrombus formation.[30] No restriction will be placed on dose or route of administration. However, studies which combine medication regimens with non-medication-based cointerventions in a single arm will be excluded. The combinations of antithrombotic agents (eg, clopidogrel with aspirin) will be analysed as separate groups to the individual agents.

## Patient and public involvement (PPI)

This research proposal was developed with assistance from the local NIHR Research Design Service and was informed by PPI work with a focus group of people with lived experience of PAD.[31] Themes emerging from the PPI group included surprise about the level of cardiovascular risk for patients with PAD and shock at the discordance in current guidelines. The PPI work highlighted that people with PAD most valued being alive without disabling complications such as stroke, heart attack or amputation, and felt that further research was important to establish the best antithrombotic agent/combination and if possible offer tailored advice for key patient groups.

## Planned further PPI

Following data extraction, a visual representation of the network for each outcome will be created. This, along with the composite outcomes from studies that are being combined, risk of bias and difference in patient demographics of each included study will be discussed at a PPI focus group. This will ascertain their perception about the quality of evidence included in the study, which by extension may inform decision-making about treatment options recommended to patients. This will occur prior to the analyses being run, thereby avoiding any potential bias from the outcomes of the NMA.

The results of the analysis are likely to be numerous and complex in their nature. A second PPI focus group will discuss the outcomes of the analysis and distill the results down to a meaningful level for patients and members of the public to interpret. This is intended to facilitate dissemination and ensure that patients with PAD can access the information to make an informed decision about the best management for them with their doctor. A summary of these PPI findings will be reported jointly with the analysis.

## Search methods

An information specialist developed a comprehensive literature search in line with the guidelines laid out in the Cochrane handbook, aiming for a high level of sensitivity.[32] The search strategy underwent peer review by the Cochrane Vascular information specialist, prior to being run.[32] Searches will be performed, in English only, on Medline, Embase, Cochrane (CENTRAL), Web of Science and Google Scholar. Searches for ongoing trials will be repeated via the WHO International Clinical Trials Registry platform search portal, with supplemental searches as recommended in the Cochrane handbook.[31] All databases will be searched from their inception. No limits will be placed on study language, although a study will be excluded if a translation is not available or feasible to arrange. The reference lists of included studies will be hand searched for further articles. All searches will be repeated prior to the end of the meta-analysis to identify any subsequently published literature before the dataset is locked.

## Selection of studies

The output of all searches will be imported into Covidence systematic review management software and screened for duplicates.[33] Once duplicates have been removed, two team members will independently screen the list of original papers by title and abstract to identify all RCTs of antithrombotic medication reporting one of the primary outcomes. For identified studies, all additional data sources will be sought using focused searches for further published articles, published letters, trial registry entries and requests for clinical study reports submitted to medical regulatory authorities. These documents will be bundled and screened in totality against the eligibility criteria independently by two blinded reviewers. The reviewers will attempt to resolve any disagreements once unblinded and a third reviewer will act as a tie-breaker where no consensus can be reached.

## Data extraction and management

Two team members will extract the data from the full text and online supplemental materials of included studies into a prepiloted data collection form in Covidence developed in collaboration with the statistician. Data on RCT design, participant baseline characteristics, study interventions, methods, all reported study outcomes, results and the authors' conclusions will be extracted and recorded as detailed in the Cochrane handbook.

Where patients with PAD represent a subgroup of the overall study population, data will be collected for both the study population as a whole and the patient with PAD subgroup where available. The details of the PAD subgroup including disease status and proportion of non-lower limb atherosclerotic patients will also be recorded. This will permit a sensitivity analysis of the likelihood that using whole-study data rather than subgroup-specific data would introduce bias within the analysis of AEs. Where multiple timepoints in follow-up are reported, the longest timepoint shall primarily be used for all outcomes, with subsequent sensitivity analyses exploring the effect of the timepoint of outcome measurement.

## Objectives

### Primary objectives

1. Define the relative efficacy and hierarchy of efficacy of all antithrombotic medication regimens, previously investigated in RCTs, at reducing the risk of major adverse cardiovascular events (MACE), in patients with PAD.
2. Define the relative efficacy and hierarchy of efficacy of all antithrombotic medication regimens, previously investigated in RCTs, at reducing the risk of major adverse limb-related events (MALE), in patients with PAD.
3. Define the relative efficacy and hierarchy of efficacy of all antithrombotic medication regimens, previously investigated in RCTs, at reducing the risk of death from any cause, in patients with PAD.
4. Define the relative risk and hierarchy of risk of all antithrombotic medication regimens, previously investigated in RCTs, of serious AEs including fatal bleeding, gastrointestinal bleeding, intracranial bleeding, severe bleeding into any major organ, bleeding requiring blood transfusion and/or return to theatre and bleeding requiring admission to hospital, in patients with PAD.

### Secondary objectives

5. Define the relative risk and hierarchy of risk of all antithrombotic medication regimens, previously investigated in RCTs, of any other recorded adverse drug effects, in patients with PAD.
6. Define the relative compliance to differing antithrombotic medication regimens, previously investigated in RCTs, in patients with PAD.
7. Explore the available data for differential efficacy and/or risks of all antithrombotic medication regimens (objectives 1–4), previously investigated in RCTs, in different subgroups including by gender, age, ethnicity, disease status (asymptomatic/IC/CLTI), conservative versus interventional management, type of interventional management, comorbidities (other disease states), in patients with PAD.
8. Explore whether any other subgroups of patient or disease characteristics are sufficiently well reported in the included RCTs to establish an analysis and, if so, conduct that subgroup analysis.
9. Establish whether the teams of RCTs that form the judgement forming segments of the networks for the primary objectives are willing to collaborate and have sufficient data to undertake future individual patient NMA.

## Outcomes

### Coprimary outcomes

#### Coprimary efficacy outcome 1

Composite of MACE, as defined in the available literature, but to include acute coronary syndrome, ischaemic stroke and cardiovascular death.

#### Coprimary efficacy outcome 2

Composite of MALE, as defined in the available literature, but to include acute limb ischaemia (and embolectomy/thrombectomy/thrombolysis), major amputation (at or above ankle) or need for peripheral revascularisation.

#### Coprimary efficacy outcome 3

All-cause mortality.

#### Coprimary safety outcome 4

Major bleeding, as defined in the available literature, but to include fatal bleeding, symptomatic bleeding in a critical area or organ, such as intracranial, intraspinal, intraocular, retroperitoneal, intra-articular or pericardial, or intramuscular with compartment syndrome, bleeding causing a fall in haemoglobin level of 20 g/L or more and/or bleeding requiring transfusion of red cells or whole blood.

### Secondary outcomes

Secondary efficacy outcomes will include individual components of MACE and MALE outcomes; cardiovascular death, acute coronary syndrome, ischaemic stroke, major amputation, acute limb ischaemia, thrombectomy/thrombolysis and need for a subsequent revascularisation procedure.

Secondary safety outcomes will include individual outcomes of fatal bleeding, gastrointestinal bleeding, intracranial bleeding, severe bleeding into any major organ, bleeding requiring blood transfusion and/or return to theatre, and bleeding requiring admission to hospital, venous thromboembolism, rash, discontinuation of assigned therapy for any reason, gastrointestinal symptoms resulting in discontinuation of assigned therapy. Additionally, any further drug-related AEs reported will be included to allow for the identification of unexpected AEs. Adherence to therapy by any quantitative measure will be included, where reported.

### Risk-of-bias assessment

Two study members with domain and meta-analysis expertise will independently assess all included trials for risk of bias using the 'Risk-of-Bias V.2' tool,[34] then areas of uncertainty in these assessments will look to be resolved by reference to collateral information sources as described above (trial registries and regulatory submissions) and contact with the primary investigators if necessary. A risk-of-bias assessment will be undertaken separately for primary outcome analyses and subgroup analyses.

### Measures of treatment effects

For all binary outcomes, the preferred data to collect will be absolute numbers of events and numbers at risk, and risk ratios will be calculated. If the absolute number of events is not available, risk or ORs with defined CIs and/or SE will be the preferred alternative. For continuous outcomes, the mean and SD of the group will be the first choice of measure.

## Unit of analysis issues

Analyses will be conducted at medication regimen level. Dose will not be primarily considered but a secondary sensitivity analysis on the primary outcomes will be undertaken to see if any dose effects are evident. It is not anticipated that any trials will report data in another manner regarding this; however, advice will be sought from the statistician should this occur.

## Missing data

Only published data will be analysed. Missing data will be considered within the risk-of-bias assessment. If data is made available through direct correspondence, it will be published as an online supplemental appendix to the review and in the studies data repository.

## Transitivity

Recruitment criteria differ between the RCTs and as such the transitivity assumption may be threatened. This is perceived to particularly be the case when considering trials of stable PAD compared with trials of periprocedural/postprocedural PAD. The validity of the transitivity assumption will be assessed quantitatively by considering the incoherence factor, which involves the comparison of direct and indirect effective estimates for each pairwise comparison in the network, where they both exist; this will be evaluated using both the local and global strategies detailed in the Cochrane handbook.[31]

## Assessment of reporting

Aspirin is expected to be the most common comparator within RCTs in this NMA, which will facilitate exploration of publication bias using comparison-adjusted funnel plots.[35] The team has considered selective reporting within trials throughout the study design phase and will qualitatively consider the effect of this during subsequent phases.

## Data synthesis

The full analysis plan is available online.[36] Studies will be summarised including the direct comparisons made, population characteristics and characteristics of those patients with PAD. The medication regimen for both overall and PAD populations will be summarised separately.

Antithrombotic regimens will be grouped into common nodes based on the drugs used in that arm. A network diagram will be constructed for each outcome, where the size of each node is proportional to the number of patients assigned to that intervention, and the thickness of each line is based on the inverse of the variance of the direct comparison. Interventions that are absent from a particular network will be highlighted.

Analysis of primary and secondary outcomes will be undertaken on a whole network basis, subject to the checks of assumptions outlined above, wherever networks can be formed based on published data. All NMAs will create pairwise comparisons of medication regimens, and a ranking of all medication regimens will be produced

with risk of bias estimates published alongside. Subgroup analyses are discussed further below.

We anticipate a complex NMA based on prior knowledge. The analysis is planned by the statistician (CH) in R Statistical Software (V.4.1),[37] primarily using the netmeta package.[38] If the transitivity assumption is violated as the result of differing RCT recruitment criteria, then separate networks will be established to allow meaningful analysis of these data.

## Subgroup analysis

The PPI group highlighted the importance of being able to provide individualised recommendations to patients. Therefore, the intent is to extract all published results for subgroups and perform all NMAs that are possible. Anticipated subgroups are PAD state (such as IC and CLTI), sex, age, ethnicity, key comorbidities and periprocedural type/status. We also intend to analyse any other subgroups that have sufficient data published but are not anticipated.

Subgroup analyses as described above, will be undertaken wherever subnetworks can be formed based on published data. The inclusion of these subanalyses will likely be susceptible to non-reporting bias as they are less likely to have formed part of the original RCT per-protocol analysis plan. Candidate interventions that are absent due to non-publication of a particular subgroup analysis will be clearly identified.

## Sensitivity analysis

Prespecified sensitivity analyses, as permitted by available data, will explore consistency in the findings regarding (1) patients with stable versus periprocedural PAD, (2) primary outcomes limited to patients with PAD only (to ascertain whether AE rates in trials of mixed atherosclerotic disease, eg, coronary heart disease and stroke, as well as PAD, are applicable to the PAD only population), (3) inclusion of studies at low risk of bias only, (4) restriction of included patients to those with IC and/or CLTI in isolation, (5) primary outcomes by dose of antithrombotic agents (variable dosing strategies, particularly of aspirin, are anticipated to affect AE rates) and (6) shorter and longer timepoints (as the optimal antithrombotic regimen may change over time for patients with PAD). Further post hoc sensitivity analyses may be developed and will be reported transparently as post hoc in the final report.

## Confidence in cumulative evidence

A summary of findings table will be constructed for the key outcomes, including MACE, MALE, all-cause mortality, major amputation as an individualised outcome and major bleeding. The underlying quality of evidence for each of these outcomes will be assessed according to the GRADE framework for NMAs, which classifies interventions by both the relative treatment effect size and certainty of evidence.[39 40]

## Ethics and dissemination

### Clinical impact

The results will be presented to clinical domain specialists at UK national and international vascular and endovascular conferences. The presentations will be made to key stakeholders including NICE, vascular and endovascular societies/charities and the all-party parliamentary group for vascular and venous disease. The lead applicants' institutions public relations team will maximise the visibility and availability of the results, both to patients and clinicians, through social and traditional media. The results will be disseminated through the UK national societies of vascular specialists and general practitioners to maximise the impact on clinical practice. The authors anticipate that the results of the research would be sufficient to trigger a review of the current NICE guidelines.

### Patient impact

The patient accessible report codeveloped with the PPI group will be submitted to UK vascular disease charities, such as the Circulation Foundation and the British Heart Foundation, for publication on their websites.

### Data storage

The data generated from this research will be an essential output. The data collected from screening, risk assessments, study results and statistical coding will be made available on the lead applicants institution research data repository to ensure ongoing future access. This will facilitate research efficiency if/when future trials become available, and the analysis requires updating. We will seek to establish a resource that can be added-to, re-evaluated or reanalysed with future trail data. A link to the data repository will be provided in the PROSPERO record for the systematic review.

### Publications

The authors anticipate that the primary paper will be published, open access, in a high-impact cardiovascular journal. Any further publications will also be submitted, open access.

**Author affiliations**
¹Molecular and Clinical Sciences Research Institute, St George's University of London, London, UK
²St George's Vascular Institute, St George's University Hospitals NHS Foundation Trust, London, UK
³Hull York Medical School, Hull, UK
⁴Department of Vacular Surgery, Hull University Teaching Hospitals NHS Trust, Hull, UK
⁵NHS Liaison Librarian, St George's University of London, London, UK

**Acknowledgements** The authors thank Candida Fenton from the Cochrane Vascular Group, University of Edinburgh, Edinburgh, UK, for peer review of the search strategy.

**Contributors** All authors drafted and revised the manuscript. INR is the guarantor for this work.

**Funding** This project is funded by the National Institute for Health and Care Research (NIHR) under its Research for Patient Benefit (RfPB) Programme (Grant Reference Number NIHR204123).

**Disclaimer** The views expressed are those of the author(s) and not necessarily those of the NIHR or the department of health and social care.

**Competing interests** None declared.

**Patient and public involvement** Patients and/or the public were involved in the design, or conduct, or reporting, or dissemination plans of this research. Refer to the Methods section for further details.

**Patient consent for publication** Not applicable.

**Provenance and peer review** Not commissioned; externally peer reviewed.

**ORCID iDs**
David Brooke Sidebottom http://orcid.org/0000-0003-3455-6659
Iain Nicholas Roy http://orcid.org/0000-0001-9701-8319

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
