## [Reviewer comments · BMJ Open]

ARTICLE DETAILS

TITLE (PROVISIONAL)	Antithrombotic drugs for cardiovascular risk reduction in patients with lower limb peripheral arterial disease: protocol for a systematic review & network meta-analysis of randomised controlled trials
AUTHORS	Sidebottom, David; Huang, Chao; Carradice, Daniel; Holt, Peter; John-Pierre, Karen; Roy, Iain

VERSION 1 – REVIEW

REVIEWER	AH Davies Imperial College London, Vascular
REVIEW RETURNED	02-Apr-2023

GENERAL COMMENTS	Excellent well put together
-----------------------------

REVIEWER	Mario Enrico Canonico University of Colorado
REVIEW RETURNED	17-Apr-2023

GENERAL COMMENTS	David B. Sidebottom et al. present the protocol for a network meta-analysis "Antithrombotic drugs for cardiovascular risk reduction in patients with lower limb peripheral arterial disease: protocol for a systematic review & network meta-analysis of randomised controlled trials". The topic is important since the new evidence in the field and the design of this meta-analysis appears appreciate. I have some minor comments: 1) The Authors said there are no RCT that compared clopidogrel vs asa + rivaroxaban. Do the Authors refer to PAD patients managed by medical therapy alone? Of course this is true but why they did not mention the comparison among DAPT (asa + clopidogrel) vs DPI (aspirin + rivaroxaban). To my knowledge this would be the right comparison in PAD patients after LER. Moreover, the influence of Clopidogrel was already assessed by VOYAGER PAD (Hiatt WR, et al. Rivaroxaban and Aspirin in Peripheral Artery Disease Lower Extremity Revascularization: Impact of Concomitant Clopidogrel on Efficacy and Safety. Circulation. 2020 Dec 8;142(23):2219-2230).2) I would suggest to add more details about the aims. Do the Authors refer their research in symptomatic PAD patients? Hence, it would be appropriate to further specify each clinical setting (e.g. patients managed by medical therapy and/or after LER). This should be specified also along the PAD population in the methods.3) About the secondary outcomes of the planned work, I would
--

	suggest to add data on antithrombotic therapy in PAD patients with concomitant CAD or carotid disease (polyvascular disease patients). This is an important point since the high ischemic and bleeding risk population as those with polyvascular disease. I would suggest to add and discuss this updated review which includes antithrombotic therapy in PAD patients in different clinical settings. Canonico ME, et al. Antithrombotic Therapy in Peripheral Artery Disease: Current Evidence and Future Directions. J. Cardiovasc. Dev. Dis. 2023, 10(4):164.
--	---

VERSION 1 – AUTHOR RESPONSE

Reviewer 1:

Thank-you for your time and comment.

Reviewer 2:

Thank-you for your time and comments. We have taken a more overarching approach to the protocol manuscript than perhaps you are suggesting. We have specified the methods by which subgroups will be identified and subsequent analyses will be performed rather than pre-empt which of the subgroup analyses will actually be possible and undertaken.

1. The Authors said there are no RCT that compared clopidogrel vs asa + rivaroxaban. Do the Authors refer to PAD patients managed by medical therapy alone? Of course this is true but why they did not mention the comparison among DAPT (asa + clopidogrel) vs DPI (aspirin + rivaroxaban). To my knowledge this would be the right comparison in PAD patients after LER. Moreover, the influence of Clopidogrel was already assessed by VOYAGER PAD (Hiatt WR, et al. Rivaroxaban and Aspirin in Peripheral Artery Disease Lower Extremity Revascularization: Impact of Concomitant Clopidogrel on Efficacy and Safety. Circulation. 2020 Dec 8;142(23):2219-2230).

Response:

The authors are not aware of any study that has or is currently randomizing patients to clopidogrel vs asa + rivaroxaban in any PAD disease state or peri-interventional state- that is looking at the specified outcomes.

The comparison of DAPT (aspirin + clopidogrel) vs aspirin + rivaroxaban is of course a relevant one in the peri-procedural (endovascular revascularisation) period due to current guidelines that advocate DAPT. However these recommendations are largely based on a single (likely underpowered) RCT study- MIRROR (Management of peripheral arterial interventions with mono Or dual antiplatelet therapy trial). As such single agent Clopidogrel may be a reasonable candidate anti-thrombotic post endovascular revascularisation.

The number of candidate interventions that have been investigated multiplied by the number of peri-procedural states (types and position of endovascular interventions + types and position of open operations) means it is not possible to address the evidence base for them all in a protocol paper. We have expanded on the secondary prevention in stable disease due this being the single largest group of patients and the group that has specific NICE guidance – which drives the UK practice which is the interest of the funder.

The influence of Clopidogrel in VOYAGER PAD quoted above is a (very interesting) non-randomised effect, clopidogrel was added to the other blinded antithrombotics at practitioner discretion therefore making particularly the adverse/harm outcomes open to bias.

2. I would suggest to add more details about the aims. Do the Authors refer their research in symptomatic PAD patients? Hence, it would be appropriate to further specify each clinical setting (e.g. patients managed by medical therapy and/or after LER). This should be specified also along the PAD population in the methods.

Response:

We have stated in the first sentence of the aims section (page 4 line 53) that the population being studied is patients with symptomatic lower limb PAD.

A subgroup analysis investigating medical therapy periprocedurally in lower limb revascularisation is already pre-specified in the protocol (page 9 line 20). We have made a minor revision to the wording to make this clearer but this is a small subgroup that from pre-existing knowledge is unlikely to have RCT evidence in all candidate interventions.

3. About the secondary outcomes of the planned work, I would suggest to add data on antithrombotic therapy in PAD patients with concomitant CAD or carotid disease (polyvascular disease patients). This is an important point since the high ischemic and bleeding risk population as those with polyvascular disease. I would suggest to add and discuss this updated review which includes antithrombotic therapy in PAD patients in different clinical settings. Canonico ME, et al. Antithrombotic Therapy in Peripheral Artery Disease: Current Evidence and Future Directions. *J. Cardiovasc. Dev. Dis.* 2023, 10(4):164.

Response:

The high prevalence of carotid artery disease in this population is important and patients with concomitant peripheral and carotid artery atherosclerotic disease have been prevalent in published randomized trials. Therefore, it is likely this will form one of the subgroups analyses, however it will only be possible if enough studies report outcomes based on carotid artery disease status. We have not specified any specific co-morbidities including even more prevalent co-morbidities such as coronary artery disease. The subgroup methods for key co-morbidities are pre-specified in the protocol (page 9 line 23-27).

The authors thank the reviewer for drawing their attention to the review article published this year and will consider the findings from this study in this context when results of the review are available.

VERSION 2 – REVIEW

REVIEWER	Mario Enrico Canonico University of Colorado
REVIEW RETURNED	13-Jun-2023
GENERAL COMMENTS	The Author addressed all of my comments.